# Two dominant boreal conifers use contrasting mechanisms to reactivate photosynthesis in the spring

Qi Yang [1,2], Nicolás E. Blanco [1,3]*, Carmen Hermida-Carrera [1], Nóra Lehotai[1], Vaughan Hurry[4]* & Åsa Strand[1]*

Boreal forests are dominated by evergreen conifers that show strongly regulated seasonal photosynthetic activity. Understanding the mechanisms behind seasonal modulation of photosynthesis is crucial for predicting how these forests will respond to changes in seasonal patterns and how this will affect their role in the terrestrial carbon cycle. We demonstrate that the two co-occurring dominant boreal conifers, Scots pine (*Pinus sylvestris L.*) and Norway spruce (*Picea abies*), use contrasting mechanisms to reactivate photosynthesis in the spring. Scots pine downregulates its capacity for $CO_2$ assimilation during winter and activates alternative electron sinks through accumulation of PGR5 and PGRL1 during early spring until the capacity for $CO_2$ assimilation is recovered. In contrast, Norway spruce lacks this ability to actively switch between different electron sinks over the year and as a consequence suffers severe photooxidative damage during the critical spring period.

[1] Umeå Plant Science Centre, Department of Plant Physiology, Umeå University, SE 901 87 Umeå, Sweden. [2] State Key Laboratory of Tree Genetics and Breeding, Chinese Academy of Forestry, Beijing 100091, China. [3] Centre of Photosynthetic and Biochemical Studies (CEFOBI-CONICET), Faculty of Biochemical Science and Pharmacy, Rosario National University, S2002LRK Rosario, Argentina. [4] Umeå Plant Science Centre, Department of Forest Genetics and Plant Physiology, Swedish University of Agricultural Sciences, SE 901 83 Umeå, Sweden. *email: blanco@cefobi-conicet.gov.ar; vaughan.hurry@slu.se; asa.strand@umu.se

Boreal forests are dominated by evergreen conifers, and the boreal climate presents a particular seasonality characterized by periods of active growth interspersed with periods of seasonal dormancy to minimize damage from severe cold. The direct consequence of this lifestyle is an environmentally regulated seasonal photosynthetic activity that at larger scales affects the global $CO_2$ budget[1–3]. The $CO_2$ photosynthetically fixed by boreal forests represents ~22% of global $CO_2$ storage by established forests, which in turn account for ~30% of the global C uptake[4]. Thus, the boreal forests are key players in balancing the global carbon cycle to reduce the impact of greenhouse emissions on future global climate. The spring recovery of photosynthesis in evergreen conifers is a crucial process for boreal forests, and proper timing of this event is a trade-off between maximizing the full growing season and minimizing damage from exposure to the combined stresses of cold temperatures and high irradiance[5]. To maintain a balance between the light energy captured to drive photosynthesis and the metabolic demand through these dynamic seasonal growth cycles, cold-tolerant plants deploy two main strategies to cope with low temperature: (i) upregulation of metabolic sink capacity[3,5] and/or (ii) downregulation of photochemical efficiency to balance the collection and usage of light energy by the photosynthetic apparatus[3]. The mechanisms of sink modulation by plants during cold acclimation have mainly been studied in herbaceous plants[6,7]. These studies show that during acclimation, cold hardy plants recover photosynthetic flux and sink capacity through a remodeling of primary carbon metabolism. However, little is known about how or whether conifers also regulate their sink capacity to protect the photosynthetic electron transport chain (PETC) at low temperatures when $CO_2$ assimilation is repressed by cold and winter dormancy[3,5]. To avoid excitation energy being in excess, plants have evolved different mechanisms to sustain a functional PETC under nonoptimal growth conditions[8,9], such as non-photochemical dissipation of excess energy as heat (NPQ)[10] and the routing of electrons to alternative pathways through alternative electron flows (AEF)[8]. AEF is composed of cyclic electron transport (CET) and pseudo-CET. At least two routes for CET are widely accepted: the PGR pathway, involving PGR5 (PROTON GRADIENT REGULATION 5) and PGRL1 (PGR5-like 1), and the NADH dehydrogenase-like complex (NDH)-mediated pathway[11,12]. However, although PGR5 has been proven to control ΔpH across the thylakoid membrane, the direct involvement of PGR5 in electron transport to plastoquinone (PQ), and therefore the impact of a PGR5/PGRL1-dependent CET pathway, is currently under debate[12–14]. In conifers, the NDH-mediated pathway is absent[15], but the mechanism for pseudo-CET is prominent and involves the flavodiiron (FLV) proteins[16]. The FLV proteins are conserved from chlorophyte algae to gymnosperms, but lost in angiosperms[17]. FLVs have been functionally evaluated by heterologous expression in Arabidopsis and rice, and shown to protect PSI against fluctuating light conditions[18,19].

In European boreal forests, Scots pine and Norway spruce are co-occurring dominant evergreen conifers. They are classified into the same functional plant group, and are therefore often expected to have similar responses to environmental change. However in natural stands, Scots pine is a pioneer species whereas Norway spruce is a late successional species[20], which indicates that their physiological plasticity in response to environmental variation may differ. To elucidate the mechanism used for photoprotection by boreal conifers during the spring recovery phase when photosynthesis is reactivated, we analyzed the photosynthetic responses in Scots pine and Norway spruce over the entire year. We found profound differences between the two species. Pine demonstrated a clear modulation of electron sink capacity over the year where the capacity for $CO_2$ assimilation was downregulated during winter and then gradually upregulated during spring in response to warming. To compensate for the reduced $CO_2$ assimilation capacity during the critical late winter–early spring months pine increased alternative electron sinks to protect the photosystems from photodamage. In contrast, Norway spruce lacks this mechanism and as a consequence suffered more severe photooxidative damage during late winter and early spring as shown by larger fluctuations in $F_v/F_m$ and increased thylakoid lipid peroxidation. Our results demonstrate that the two co-occurring dominant boreal conifers use contrasting mechanisms to reactivate photosynthesis in the spring.

## Results

**Seasonal changes in photosynthetic performance.** To investigate the seasonal photosynthetic performance of the two boreal key species, Scots pine and Norway spruce, the photosynthetic capacity and functionality was investigated over a full year from mature (80 + year old) trees growing together in a mixed coniferous forest in northern Sweden (64° 00′ 21.24″N, 19° 54′ 00.24″E) (Fig. 1). Ambient temperature during this time period varied by 50 °C (Fig. 1a). Measurements of chlorophyll fluorescence demonstrated a clear seasonal pattern of photosynthetic performance where both species maintained a fully functional PETC, with a maximum quantum yield of PSII ($F_v/F_m$) > 0.8, for only 5 months (June–October) of the year (Fig. 1b). During the winter period, PSII is inactivated with both species showing a decline in $F_v/F_m$ beginning in late October, reaching low values of $F_v/F_m$ of 0.38 through to the end of March (Fig. 1b). During spring recovery (April–May), $F_v/F_m$ increased in both species but, especially in Norway spruce, large variations in the $F_v/F_m$ values were observed where high values were followed by a sharp drop, indicative of photodamage associated with frost events. Scots pine demonstrated a slower and less volatile recovery of $F_v/F_m$, which in turn supported higher rates of electron transport of PSII (ETR(II)), and in particular by very high rates of electron transport by PSI (ETR(I)) during the critical March to May period of spring reactivation (Fig. 1c, d). The large variation in PSI activity over the year observed in Scots pine was not shown in Norway spruce (Fig. 1c, d).

**Norway spruce chloroplasts are more sensitive to photodamage.** Analysis of the chloroplast ultrastructure during the winter-to-spring transition showed that between winter (February) and early spring (March), chloroplasts showed an almost complete loss of grana structures (Fig. 2a; Supplementary Fig. 1). Quantification of the TEM images revealed that during spring recovery (March), Norway spruce underwent more extensive remodeling of the thylakoid membranes compared with Scots pine, possessing fewer grana stacks per chloroplasts (19 ± 4 (mean ± SD, n = 6)) than in Scots pine (31 ± 9) and reduced effective surface per granum (2.4 ± 0.3 and 3.2 ± 0.4 thylakoid membranes/granum for Norway spruce and Scots pine, respectively) (Fig. 2a). In addition, Norway spruce demonstrate larger numbers of plastoglobuli (Fig. 2b, c) and higher levels of malondialdehyde (MDA) (Fig. 2d), a by-product of lipid peroxidation, compared with Scots pine. Taken together, the data suggest Norway spruce suffers greater oxidative stress during the spring recovery phase compared with Scots pine, as also indicated by the large volatility in $F_v/F_m$ observed in Norway spruce during the spring period (Fig. 1b).

**Seasonal and temperature responses of $CO_2$ assimilation.** Scots pine demonstrated variable rates of ETR(I) over the year, with particularly high rates during the critical spring reactivation period (Fig. 1d). Scots pine also showed a large variation in photosynthetic $CO_2$ assimilation capacity under saturating conditions over the year. The light- and $CO_2$-saturated $CO_2$

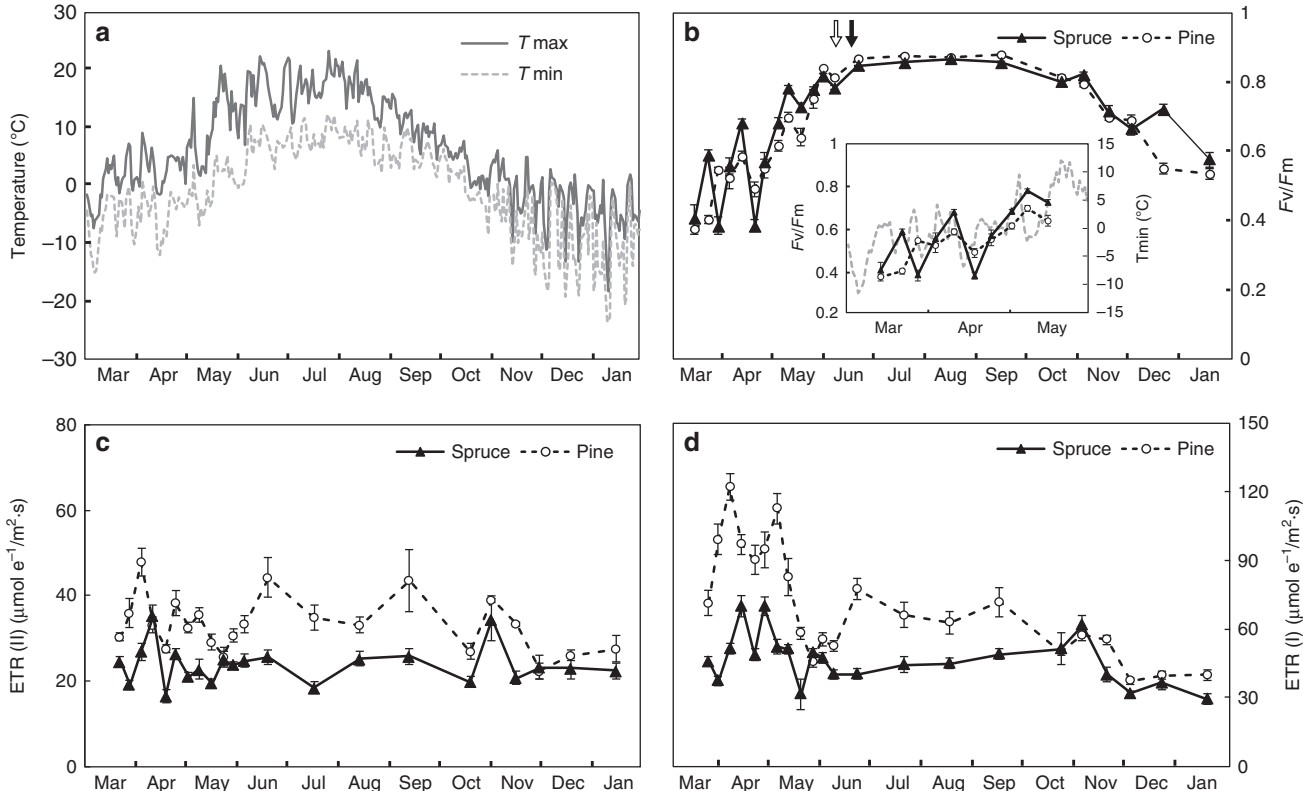

**Fig. 1 Scots pine and Norway spruce demonstrate clear differences in the capacity for photosynthetic performance during the transition period from winter to summer.** Two mature conifer trees, *Pinus sylvestris* (Scots pine) and *Picea abies* (Norway spruce), located near Vännäs, Sweden, were selected for analysis. Needles on south-facing branches that were developed in 2016 were used for the analysis. Needles of Scots pine (open circle) and Norway spruce (closed triangle) were collected during March 2017 through January 2018. **a** Air temperature of daily maximal (solid line) and minimal (dash line) temperature for the period February 2017 to January 2018. **b** $F_v/F_m$, maximum quantum efficiency of photosystem II (PSII) photochemistry. Data collected from March to May are shown in a close-up view. The dates of budburst in Scots Pine and Norway spruce are indicated with open and closed arrows, respectively. **c**, **d** ETR(II) and ETR(I), electron transport rate through PSII and through PSI, respectively. Data were collected under actinic light intensity of 1292 μmol photons $m^{-2}\,s^{-1}$. Each data point represents mean of six biological replicates (mean ± SE, $n = 6$).

assimilation rate was low in needles collected during the winter and spring months, and then increased dramatically toward the end of May once $T_{min}$ rose above 0 °C in the field (Fig. 1), to remain high during the summer months, and then decline in October to again reach low flux capacities in early winter (Fig. 3a). Similar trends were also observed for the $CO_2$ assimilation rates at ambient $CO_2$ concentrations (Supplementary Fig. 2). This response pattern in Scots pine is similar to what has been reported from field measurements[21,22] and indicates that the low fluxes measured during autumn and winter are not only a result of the inhibition of photosynthesis by low temperature but are also due to a downregulation of $CO_2$ assimilation capacity during the autumn and winter months. In contrast to Scots pine, no seasonal variation in the maximal capacity for $CO_2$ assimilation was observed in Norway spruce (Fig. 3a), although field measurements show seasonal variation in $CO_2$ assimilation reflecting low-temperature inhibition of $CO_2$ assimilation[23]. Supporting this apparent differential regulation of $CO_2$ assimilation capacity by the two species, a controlled recovery experiment with field samples collected in April demonstrated a clear recovery of the $CO_2$ assimilation capacity and in the maximal rates of carboxylation ($V_{cmax}$) and of electron transport ($J_{max}$) in warming-recovered Scots pine, but very little difference between cold-acclimated field and warming-recovered needles of Norway spruce (Fig. 3b, e).

A climate chamber experiment where cold-acclimated seedlings were transferred to warm temperature (22 °C) showed that the net rate of $CO_2$ assimilation and $V_{cmax}$ increased significantly in Scots pine following exposure to warm temperature, whereas in Norway spruce no significant difference was observed between the cold-acclimated and warm-shifted samples (Fig. 3c, e). Supporting these differences in seasonal $CO_2$ assimilation, an increase in *rbcL* (gene encoding RuBisCO large subunit) expression was observed in Scots pine during the late spring (Fig. 3d). No increase in *rbcL* expression was shown in Norway spruce during the winter–spring transition period (Fig. 3d). Starch, which was only observed to accumulate after photosynthesis was activated (Supplementary Fig. 3), began to accumulate as early as April in Norway spruce, but was not detected in Scots pine until later in May (Supplementary Fig. 3). Additional climate chamber experiments, where either temperature or day length was gradually increased from 4 °C and 4 h light to 22 °C and 22 h light, respectively, demonstrated that the recovery of $CO_2$ assimilation activity in Scots pine is controlled solely by increased temperature, and not by increased day length (Supplementary Fig. 4).

**PSI activity is essential during the winter–spring transition.** During exposure to high levels of excitation energy at low-temperatures plants are prone to PSI acceptor-side limitation, which eventually leads to PSI photodamage[24–26], from which plants recover more slowly compared with recovery from photodamage to PSII. Furthermore, PSI photoinhibition is believed to

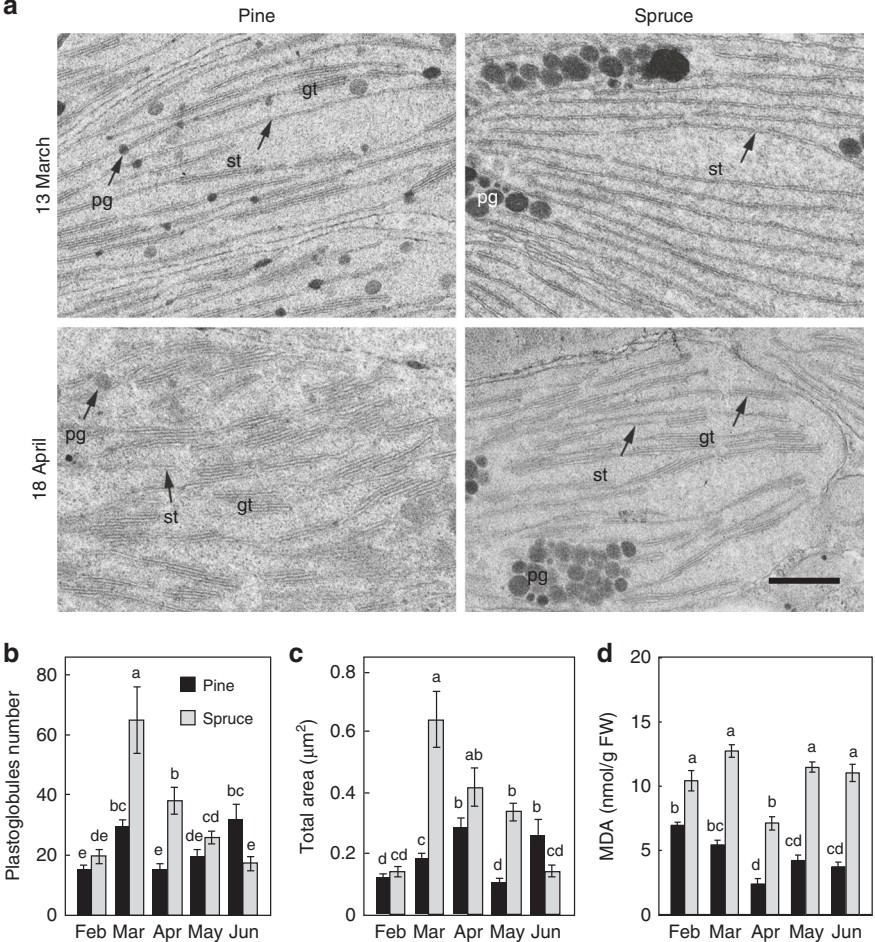

**Fig. 2 The chloroplasts in Norway spruce are more sensitive to photodamage during the spring recovery period compared with the chloroplasts in Scots pine. a** Transmission electron micrographs (TEM) of chloroplast structures in needles from Scots pine and Norway spruce collected in the field during March and April. Representative images are shown. Bar: 0.5 μm. gt grana thylakoid (stacked), st stromal thylakoid (unstacked), pg plastoglobulus. The TEM images from the remaining months are found in Supplementary Fig. S1. The number (**b**) and total area (**c**) of plastoglobules per chloroplast from the transmission electron micrographs from Scots pine (black) and Norway spruce (gray, mean ± SE, $n = 8$–12). **d** Malondialdehyde (MDA) content in Scots pine (black) and Norway spruce (gray) needles. Samples were collected from February to June, and each data point represents the mean of four replicates (mean ± SE, $n = 4$). Significant differences were indicated with different letters above the bars (one-way ANOVA, $P < 0.05$).

have more severe consequences for plant metabolism compared with PSII photoinhibition[27,28], making the avoidance of damage to PSI particularly important. The maintenance of an increased ETR(I) activity in Scots pine during the spring recovery phase (Fig. 1) suggests winter acclimation has led to some change in the redox poise of the PETC in Scots pine, but not in Norway spruce. Alternative electron flows (AEF) around PSI have been proposed as alternative electron pathways that can function to minimize the risk of overreduction of the PETC and damage to PSI. The relative quantum yield of AEF (Y(AEF))[29] was calculated during the spring period, representing the Δ flow between PSI and PSII contributed by CET and pseudo-CET (Fig. 4a). It is clear that in Scots pine Y(AEF) is significantly elevated during the critical spring period and then reduced during the summer months. Norway spruce, on the other hand, showed very little variation in Y(AEF) between the spring and summer months (Fig. 4a). In the controlled recovery experiment with field samples collected in April, a reduction in Y(AEF) was observed following 24 h in warm temperatures in Scots pine (Fig. 4b; Supplementary Fig. 5). No difference was observed in Norway spruce (Fig. 4b). The reduced Y(AEF) following warm exposure in Scots pine correlated with the recovery of the $CO_2$ assimilation capacity observed in the warming-recovered samples (Fig. 3b).

To test if AEF could protect PSI from photoinhibitory spring conditions in conifers, we used a dynamic saturating pulse protocol including periods of high-light (Supplementary Fig. 6) designed to mimic natural fluctuating light (FL) conditions that are the main cause of PSI photoinhibition[12,19]. We evaluated samples collected from the field in early spring (April) and from greenhouse-grown plants (control). The greenhouse grown control seedlings presented similar Y(AEF) values to their summer field samples (Fig. 4a; Supplementary Fig. 7). During the high-light treatment, the field samples collected from Scots pine showed the highest Y(I), and Norway spruce the lowest Y(I), of all four sample types measured (Fig. 4d). This high-flux capacity of PSI shown in the April samples from Scots pine was supported by the ETR(I) data (Fig. 4g). Consistent with the data from the fluctuating light experiment, ETR(I) values under saturating constant irradiances were also higher in the early spring Scots pine samples from the field compared with greenhouse-grown Scots pine samples, whereas the early spring Norway spruce samples from the field had lower rates of ETR(I) compared with their greenhouse controls (Supplementary Fig. 7). Furthermore, in Scots pine, there was no sign of an acceptor-side limitation during the high-light treatment, indicating that PSI in the Scots pine spring needles was almost fully oxidized

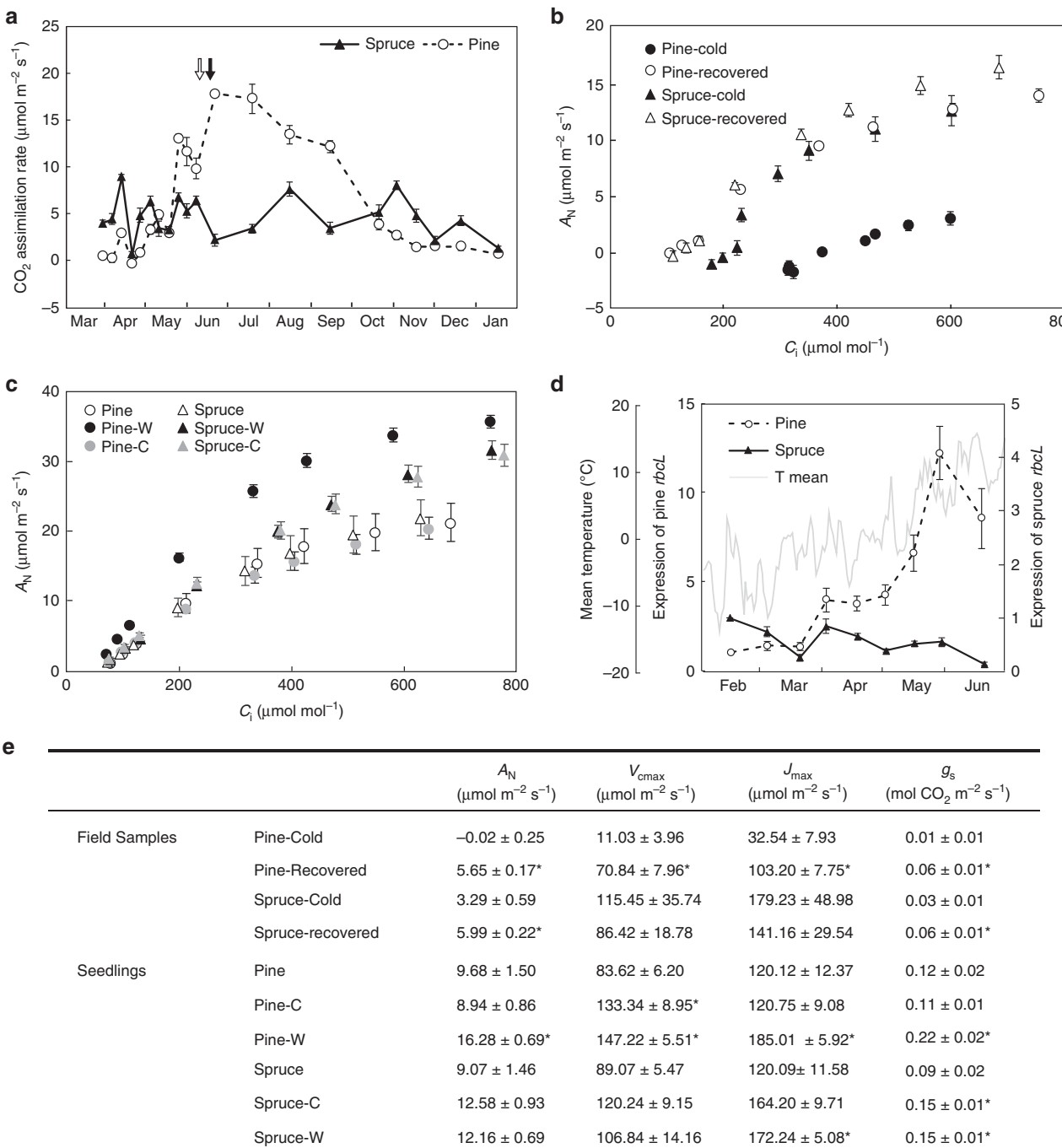

| | | $A_N$ (µmol m$^{-2}$ s$^{-1}$) | $V_{cmax}$ (µmol m$^{-2}$ s$^{-1}$) | $J_{max}$ (µmol m$^{-2}$ s$^{-1}$) | $g_s$ (mol CO$_2$ m$^{-2}$ s$^{-1}$) |
|---|---|---|---|---|---|
| Field Samples | Pine-Cold | −0.02 ± 0.25 | 11.03 ± 3.96 | 32.54 ± 7.93 | 0.01 ± 0.01 |
| | Pine-Recovered | 5.65 ± 0.17* | 70.84 ± 7.96* | 103.20 ± 7.75* | 0.06 ± 0.01* |
| | Spruce-Cold | 3.29 ± 0.59 | 115.45 ± 35.74 | 179.23 ± 48.98 | 0.03 ± 0.01 |
| | Spruce-recovered | 5.99 ± 0.22* | 86.42 ± 18.78 | 141.16 ± 29.54 | 0.06 ± 0.01* |
| Seedlings | Pine | 9.68 ± 1.50 | 83.62 ± 6.20 | 120.12 ± 12.37 | 0.12 ± 0.02 |
| | Pine-C | 8.94 ± 0.86 | 133.34 ± 8.95* | 120.75 ± 9.08 | 0.11 ± 0.01 |
| | Pine-W | 16.28 ± 0.69* | 147.22 ± 5.51* | 185.01 ± 5.92* | 0.22 ± 0.02* |
| | Spruce | 9.07 ± 1.46 | 89.07 ± 5.47 | 120.09± 11.58 | 0.09 ± 0.02 |
| | Spruce-C | 12.58 ± 0.93 | 120.24 ± 9.15 | 164.20 ± 9.71 | 0.15 ± 0.01* |
| | Spruce-W | 12.16 ± 0.69 | 106.84 ± 14.16 | 172.24 ± 5.08* | 0.15 ± 0.01* |

**Fig. 3 The capacity for CO$_2$ assimilation is downregulated in winter and reactivated with warm temperature in Scots pine. a** Seasonal photosynthetic CO$_2$ assimilation ($A_N$) was measured under saturated conditions (light intensity, 1200 µmol photons m$^{-2}$ s$^{-1}$ and CO$_2$ concentration, 800 µmol mol$^{-1}$, 23 °C). Needles of Scots pine (open circle) and Norway spruce (closed triangle) were collected from March 2017 through January 2018. The dates of budburst in Scots Pine and Norway spruce are indicated with open and closed arrows, respectively. **b** Response of assimilation ($A_N$) to the internal CO$_2$ concentration ($C_i$) in a controlled recovery experiment. Samples of Scots pine (closed circle) and Norway spruce (closed triangle) were collected from the field in April. After the initial measurements, the Scots pine (open circle) and Norway spruce (open triangle) samples were recovered in room temperature for 24 h, and measured again. **c** Response of assimilation ($A_N$) to the internal CO$_2$ concentration ($C_i$) in seedlings grown in climate chambers. The $A/C_i$ curves were measured in 1-year-old cold-acclimated Scots pine (open circle) and Norway spruce (open triangle) seedlings. Seedlings were transferred either to warm (22 °C, indicated in black) or cold (5 °C, indicated in gray) chambers for 4 weeks, and $A/C_i$ curves were determined again. **d** *rbcL* expression during the spring recovery phase in Scots pine (open circle) and Norway spruce (closed triangle). Relative expression values were normalized against the reference gene *PP2A* and related to the amount present in the February samples. Each data point represents the mean (± SE) of at least three replicates. The daily mean of air temperature (gray line) for the period February to May 2017 is shown in light gray. **e** Photosynthetic parameters calculated from the $A/C_i$ curve are shown in **b** and **c**. $A_N$ rate of CO$_2$ assimilation, $V_{cmax}$ maximum rate of carboxylation, $J_{max}$ maximum rate of electron transport, $g_s$ stomatal conductance. Units are µmol m$^{-2}$ s$^{-1}$. Asterisks indicate the significant difference between values calculated from treatments and control (one-way ANOVA, $P < 0.01$). Each data represent mean of 4–6 replicates (mean ± SE, $n = 4$–6).

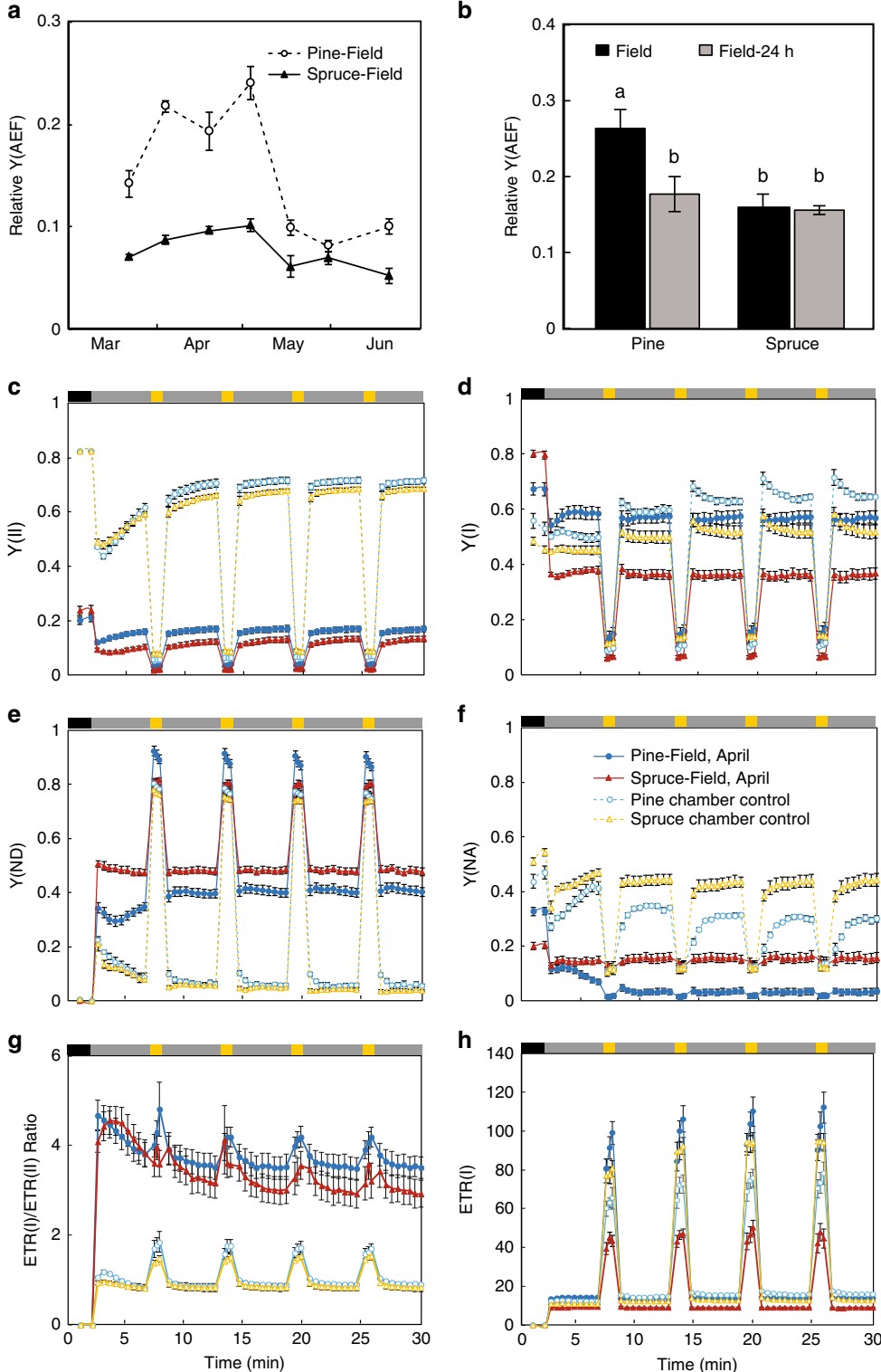

throughout the entire experiment (Fig. 4f). This is despite the reduced $CO_2$ assimilation capacity in the Scots pine spring needles (Fig. 3), suggesting AEF might have been activated as an alternative electron sink.

**The role of AEF and PGR5/PGRL1 during critical spring months.** The photosynthetic parameters indicate that AEF could contribute to photoprotection of PSI during the spring recovery phase, especially in Scots pine. The abundance of PGR5, PGRL1,

and FLVB were determined by western blot analysis in Scots pine and Norway spruce during the winter–summer transition period (Fig. 5a). Scots pine showed increasing amounts for both PGR5 and PGRL1 proteins in spring with the highest amounts present in the needles in March and April (Fig. 5a, b). In contrast, Norway spruce showed no change in PGR5 and PGRL1 protein levels in spring compared with winter or summer (Fig. 5a, b). The increase in PGR5 protein in Scots pine during the spring months correlated with an increase in *PGR5* expression (Supplementary Fig. 8). As shown previously, the PGRL1 protein exists in a

**Fig. 4 Large electron sinks downstream of PSI protect Scots pine and Norway spruce under illumination mimicking fluctuating growth light condition. a, b** The relative yield of AEF (Y(AEF)), representing the Δ flow in conifers between PSI and PSII mainly contributed by CET and pseudo-CET. Y(AEF) is calculated as Y(AEF) = Y(I) − Y(II). The yield of PSII and PSI were measured simultaneously with rapid light curves, and the values under moderate high light (536 μmol photons $m^{-2} s^{-1}$) presented here. Needles of Scots pine (open circle) and Norway spruce (closed triangle) were collected during March to June 2017 in (**a**). Samples in (**b**) were collected from the field in April, and measured immediately (black) and following recovery in room temperature for 24 h (gray). **c–h** Photosynthetic fluorescence measurement of Scots pine and Norway spruce under illumination mimicking fluctuating growth light condition. In vivo fluorescence and P700 signals monitored under 2 min dark, followed by four cycles of 5 min low light (58 μmol photons $m^{-2} s^{-1}$, gray bar) and 1 min high light (1599 μmol photons $m^{-2} s^{-1}$, yellow bar). Fluorescence parameters: **c** Y(II), operating efficiency of PSII; **d** Y(I), operating efficiency of PSI; **e** Y(ND), quantum yield of non-photochemical energy dissipation in PSI reaction centers that are limited due to a shortage of electrons (donor-side limitation); **f** Y(NA), quantum yield of non-photochemical energy dissipation in PSI reaction center s that are limited due to shortage of electron acceptors (acceptor-side limitation); **g** ETR(I)/ETR(II), the ratio of the electron transport rate of PSI to PSII; **h** ETR(I), the electron transport rate of PSI. Sun-acclimated needles from Scots pine (closed circle) and Norway spruce (closed triangle) were collected from the field in April. Seedlings from Scots pine (open circle) and Norway spruce (open triangle) grown in growth chamber (22 °C, 150 μmol photons $m^{-2} s^{-1}$ and 8/16 h light/dark cycle) were used as controls. Significant differences are indicated with different letters above the bars (one-way ANOVA, $P < 0.05$). Each data point represents the mean of 4–6 biological replicates (mean ± SE, $n = 4$–6).

reduced and oxidized form in the thylakoids[30]. The change of PGRL1 protein abundance was primarily seen for the oxidized form of PGRL1 (Fig. 5a, b). In contrast, FLVB protein levels showed no difference in accumulation during February to June (Fig. 5a, b).

The accumulation of PGR5 observed in Scots pine might constitute a safety valve for the avoidance of redox imbalance around PSI and the deleterious effects of excess excitation energy during spring. To evaluate this hypothesis, two complementary experiments were performed; First, the photosynthetic performance of field samples of Scots pine and Norway spruce collected in spring and summer was evaluated using the FL protocol described above (Fig. 5c). Scots pine spring samples show minimal evidence of acceptor-side limitations (Fig. 5c), whereas the Scots pine summer samples show a shift in the limitation of electrons from the donor side to the acceptor side during the high-light periods (decreased Y(ND) and increased Y(NA)) was observed. This shift was not observed in Norway spruce (Fig. 5c), suggesting that the change in PSI redox status in Scots pine during the spring-to-summer transition may be linked to the changes in PGR5 abundance. To confirm the contribution of PGR5, the second experiment was performed where the dynamic FL protocol was combined with antimycin A (AA) treatment, which has been extensively characterized as an inhibitor of PGR5-dependent CET[31]. Samples from Scots pine and Norway spruce were collected in the field in April, and Y(I) was determined following AA treatment and following a recovery period of 24 h at room temperature (Fig. 5d). A clear decrease in operating efficiency of PSI (Y(I)) was shown during high-intensity illumination in both Scots pine and Norway spruce treated with AA compared with the untreated control, indicating that PGR5-dependent AEF is active in the field samples of both species. However, following 24 h recovery at room temperature there was no effect of the AA treatment on Y(I) in the Scots pine needles. In contrast, the AA-mediated inhibition was still observed following warm recovery in Norway spruce (Fig. 5d). This indicates that it is only in Scots pine that the AA-sensitive activity is rapidly reversible. Furthermore, this reversibility was correlated with a rapid reduction in PGR5 protein abundance (Fig. 5e) and a recovery of $CO_2$ assimilation capacity (Fig. 3b). This feature was also observed in AA-treated Scots pine seedlings grown in cold and warm climate chambers, supporting a contribution of PGR5 to Y(I) and operating efficiency of PSII (Y(II)) when the $CO_2$ capacity is downregulated (Fig. 3c; Supplementary Fig. 9). Taken together, the data support the conclusion that PGR5 acts as a safety valve for photoprotection during the sensitive spring period in Scots pine. However, as the capacity for $CO_2$ assimilation is recovered in response to increasing temperature during spring,

and $CO_2$ assimilation can again act as the main electron sink, the PGR5 pathway is downregulated. Norway spruce, on the other hand, appears to lack this ability to actively switch between different electron sinks over the year.

## Discussion
The boreal biome is characterized by periods of active growth interspersed with periods of dormancy, resulting in strongly regulated seasonal photosynthetic activity. Our results demonstrate that the two dominant tree species of the boreal biome (Scots pine and Norway spruce) do not regulate their seasonal photosynthesis in the same way. Scots pine downregulates its capacity for $CO_2$ assimilation during winter, induces very high rates of ETR(I) and accumulates PGR5 and PGRL1 during late winter and early spring as a temporary alternative electron sink (Fig. 6). In the late spring and early summer when temperatures rise, Scots pine recovers its capacity for $CO_2$ assimilation and this is coordinated with a re-poising of the rates of ETR(II) (Supplementary Fig. 10) and ETR(I) and reduced amounts of PGR5 (Fig. 6). However, in Norway spruce the capacity for AEF and $CO_2$ assimilation is constant throughout the year and when the photosynthetic apparatus is challenged during the late winter–early spring months, when cold temperatures are experienced in combination with high irradiance, Norway spruce suffers severe photooxidative damage as shown by large fluctuations in $F_v/F_m$ and increased thylakoid lipid peroxidation (Figs. 1, 2).

Evergreen species such as Scots pine and Norway spruce maintain their foliage year-round and they therefore must possess efficient photoprotective mechanisms to cope with excess energy absorbed by a functional PETC during winter and early spring, when general metabolism (including $CO_2$ assimilation) is strongly inhibited by cold temperatures. It has been suggested that a key factor explaining how gymnosperms outcompete angiosperms at high latitudes is their ability to maintain a highly functional PSI that is able to dissipate the excess light energy captured at cold temperatures[5]. The recent finding that FLV functions as electron acceptor and can reduce $O_2$ and protect PSI from damage in gymnosperms suggests that this may be one of the key mechanisms facilitating the evergreen lifestyle of boreal conifers[32–34]. Angiosperms lack FLV[34,35] and heterologous expression of FLVs in angiosperms have demonstrated complementary functions of the different AEF components[18,19]. However, those measurements were performed on material grown under optimal conditions in climate chambers, similar to the summer samples in our experiments (Figs. 1, 3, 5), and it remains unclear whether FLV and PGR5 could function synergistically under natural physiological stress conditions such as the critical spring recovery

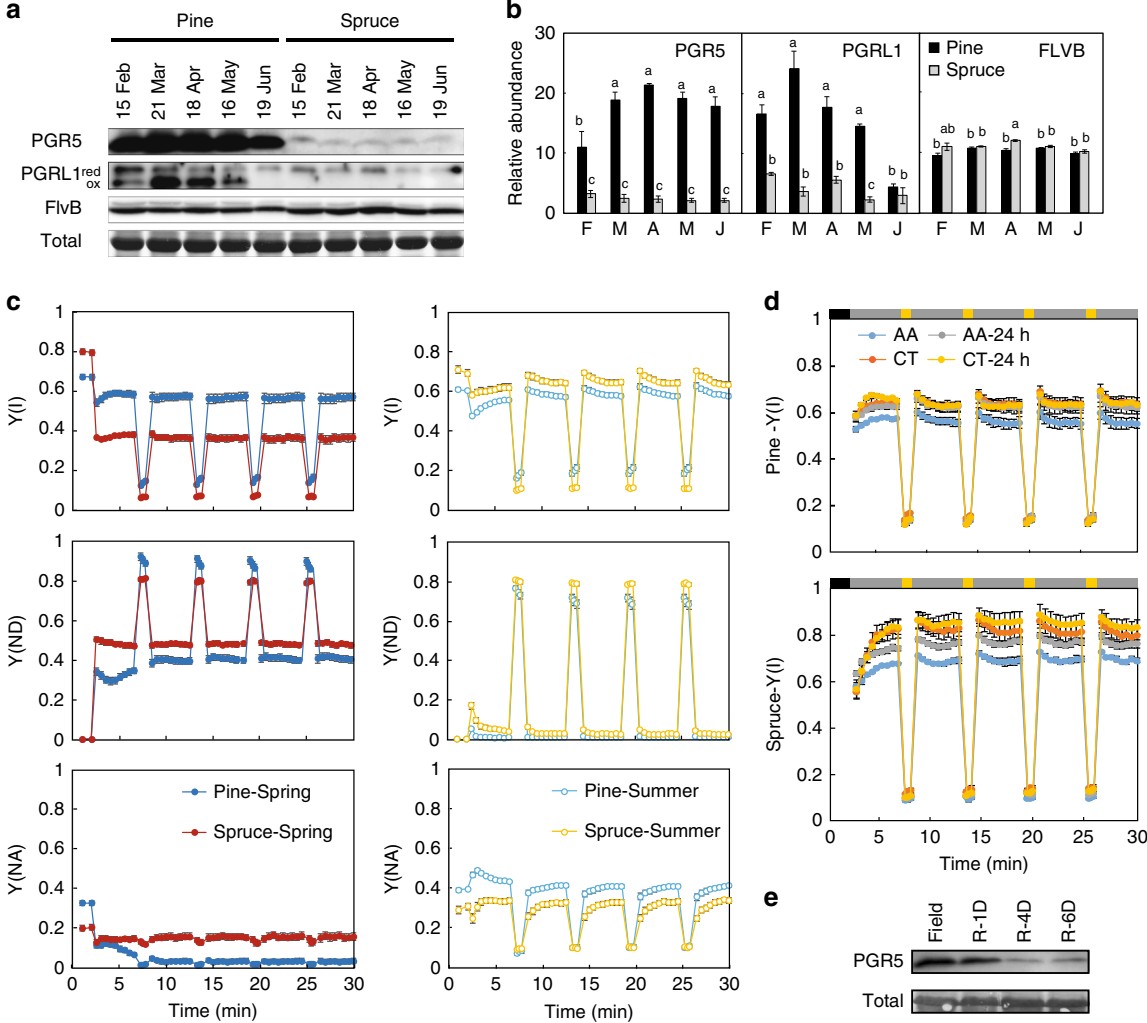

**Fig. 5 Contribution of PGR5 to photoprotection in Scots pine and Norway spruce during the critical spring months. a, b** Abundance of PGR5, PGRL1, and FLVB. Protein levels were examined by immunoblot analyses with specific antibodies against PGR5, PGRL1, and FLVB. Both reduced (red) and oxidized (ox) forms of PGRL1 protein, were detected in Scots pine, were collected from February to June 2017. Twenty-five μg of total protein was loaded per lane, and a representative band from the stained gel is shown as a loading control. Protein levels in Scots pine (black) and Norway spruce (gray) were quantified from three independent experiments using the program ImageJ. Significant differences were indicated with different letters above the bars (one-way ANOVA, $P < 0.05$). **c** PSI redox status comparison between spring and summer. The same protocol applied in Fig. 4 was used for samples of Scots pine (dark blue and light blue) and Norway spruce (red and yellow) collected from the field in April (Spring) and in June (Summer). **d** Effect of antimycin A (AA) treatment upon light fluctuations in a controlled recovery experiment. Scots pine and Norway spruce samples collected from the field in April. After the initial measurements, samples were allowed to recover at room temperature for 24 h, and measured again. Needles were treated with water (orange and yellow circles) or 200 μmol AA (blue and gray circles). The same protocol applied in Fig. 4 was used for the measurements. The parameter Y(I), operating efficiency of PSI is shown. Each data point represents the mean of four biological replicates (mean ± SE, $n = 4$). **e** Scots pine samples collected from the field in April and recovered at room temperature for 1 (R-1D), 4 (R-4D), and 6 days (R-6D) were used for determining abundance of PGR5. Representative bands from the ponceau-S stained membranes are shown as loading controls.

period. In our experiments we show that Scots pine, but not Norway spruce, resolves the winter evergreen dilemma using both PGR5 and FLVs as AEFs through an elegant and dynamic control of the PGR5 levels. As part of the reactivation of photosynthesis during early spring, Scots pine induces the expression of PGR5 (Fig. 5; Supplementary Fig. 7). This adaptive strategy combines the FLV-mediated electron sink with induced protection of both sides of PSI through the PGR5-dependent mechanism[35]. This regulatory mechanism, activated in early spring in the Scots pine needles (Fig. 6), results in increased tolerance to photoinhibition by minimizing acceptor-side limitations on PSI (i.e., photo-inhibitory PSI overreduction) when needles are exposed to periods of high-light irradiance at low temperatures. Our experiments conducted on field samples (Fig. 4) revealed that AEF in Scots

pine needles are able to control the electron transfer to PSI (high Y(ND)) and thermally dissipate the excess excitation energy (Fig. 4d). Thus, although FLV may be the dominate component of AEF during periods of productive growth, the significant decay of PSI capacity in response to AA-inhibition in the Scots pine spring needles, which is lost once $CO_2$ assimilation is recovered (Fig. 5d), suggests PGR5 plays a key role as an inducible temporary electron sink during the critical spring recovery phase in Scots pine.

The main constraint on the growth of the boreal forests is temperature-dependent season length[21,36], and ongoing climate change has caused a mean annual temperature increase of 1.5 °C in high-latitude boreal forests, driving earlier bud flush[37,38] at a time when plants remain at risk of exposure to freezing events[39].

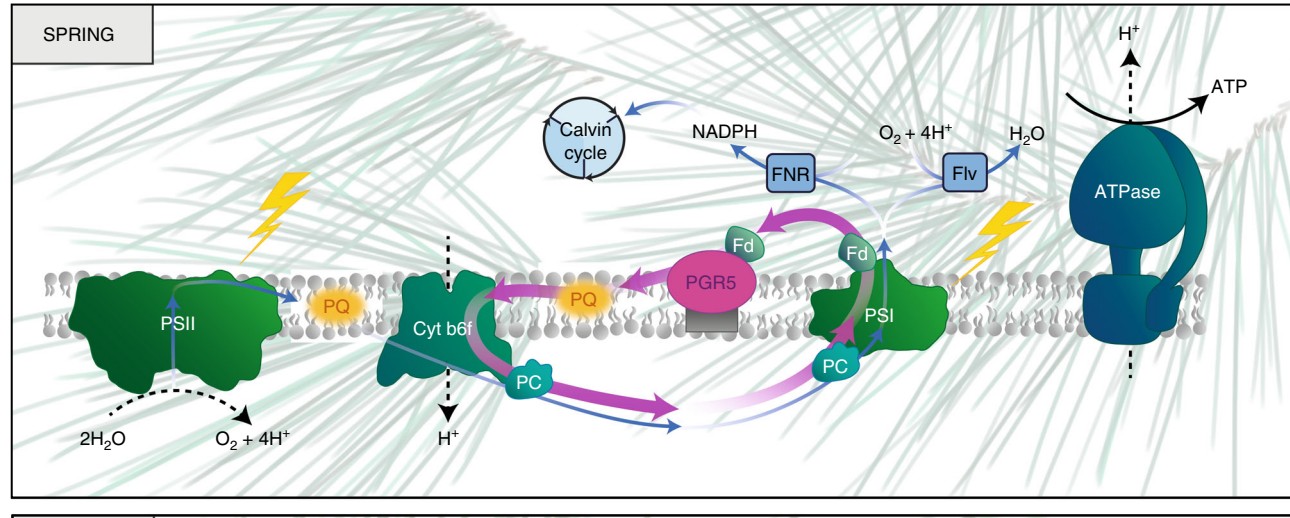

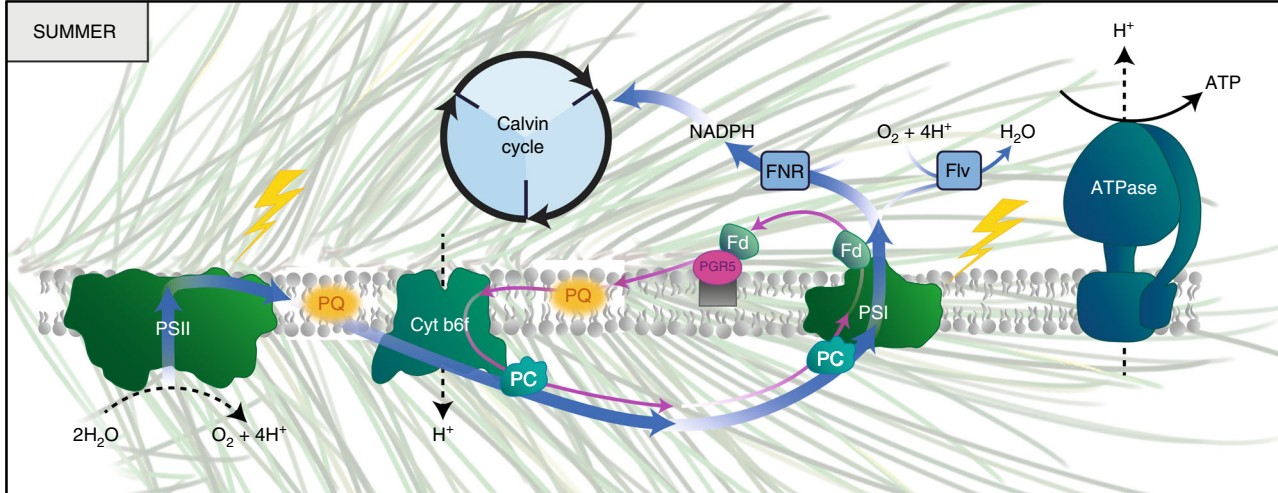

**Fig. 6 Scots pine has evolved the superior strategy for photosynthesis in the North.** Scots pine demonstrated a clear modulation of electron sink capacity over the year where the capacity for $CO_2$ assimilation was downregulated during winter and then gradually upregulated during spring in response to warming. To compensate for the reduced $CO_2$ assimilation capacity during the critical late winter–early spring months, Scots pine induces PGR5-dependent CET activity to provide an alternative electron sink to protect the photosystems from photodamage at a time when $CO_2$ assimilation capacity is limited. PC plastocyanin. Blue arrows represent linear electron transport, magenta arrow represents alternative electron transport.

Recent observations made in boreal forest biomes over Fennoscandia, North America, and Russia have identified accelerated growth in response to longer growing seasons[40–42], with the earlier arrival of spring shown to yield the greater growth benefits[43,44]. Our results show that two of the dominant conifer species utilize fundamentally different mechanisms to manage spring recovery of photosynthetic activity. This difference may in part reflect the divergent positions these two species occupy in the ecosystem, with Scots pine being an early pioneer species whereas Norway spruce is a shade tolerant late successional species that develops under a covering canopy[20]. Norway spruce and Scots pine also have different shoot and canopy structures, resulting in greater self-shading both within the shoot and within the canopy of Norway spruce plants, even in mature Norway spruce trees that have emerged from their sheltering overstory canopy. These two factors might indicate that Norway spruce has less of a need for such protective mechanisms in spring compared with Scots pine. However, we show that mature exposed Norway spruce canopies do suffer wider fluctuations in $F_v/F_m$ during spring recovery, and therefore suffer more repeat damage to the reactivating photosynthetic apparatus than Scots pine. With climate-driven earlier bud flush and an increase in frequency of spring backlash events[45], Norway spruce is likely to be more vulnerable

to spring frost damage to their canopy in the coming years. It has also recently been shown that Scots pine is more able than Norway spruce to acclimate photosynthesis and respiration to both increased seasonal temperatures and elevated $CO_2$[46]. Scots pine increased growth in response to temperature increases as high as +8 °C, whereas Norway spruce showed minimal capacity to acclimate energy metabolism and suffered growth losses at elevated seasonal temperatures[46]. These findings, together with those we report here, indicate that the pioneer species Scots pine may generally have a greater capacity to cope with environmental fluctuations and challenges than the more ecologically conservative late successional species Norway spruce. Elucidating the divergent mechanisms utilized by the dominant species in these forests will be crucial for predicting how they will respond to future changes in the timing of spring arrival. The differential responses of these two dominant species need to be accounted for in the estimations of carbon sequestration by the boreal forests, particularly in continental climates.

## Methods

**Plant material and growth conditions.** Two mature conifer trees, *Pinus sylvestris Linn.* (Scots pine) and *Picea abies (L.) Karst.* (Norway spruce), located near Vännäs, Umeå, Sweden (63° 54′ 24.34“N, 19° 45′ 25.63“E), were selected for

analysis. Air temperature at the location was monitored every day. The dates of budburst in Scots Pine (average apical shoot length reached 20 mm) and Norway spruce (average Krutzsch index 3, which represents the budburst stage) are on June 8th and June 15th, 2017, respectively[47]. To characterize seasonal photosynthetic activity, needles on south-facing branches that developed in 2016 were collected during February 2017 through January 2018. Branches were also collected from the field in April 2018 and 2019, and kept at room temperature for 24 h, 4 and 6 days for controlled recovery experiments. All experiments were performed with needle samples collected from the field unless specified. For the growth chamber experiments, 1-year old seedlings of *P. sylvestris* and *P. abies* were grown in soil in 1 L pots with a photoperiod of 8 h light/16 h dark at an irradiance of 150 μmol photons $m^{-2} s^{-1}$ and under cold (5 °C) or warm (22 °C) temperatures as indicated. For additional climate chamber experiments, the cold-acclimated seedlings were grown under gradually increasing temperature from 4 °C to 22 °C (increase 1 °C per day) with 8 h light/16 h dark or under gradually increasing day length from 4 h to 22 h light (increase 1 h light per 2 days) with 5 °C.

**In vivo chlorophyll fluorescence, P700 measurement**. In vivo chlorophyll a fluorescence and signal from oxidized P700 were monitored simultaneously with a Dual PAM-100 fluorometer (Heinz Walz Gmbh, Effeltrich, Germany) at room temperature. Needles from mature *P. sylvestris* and *P. abies* trees were dark acclimated for 30 min and then bundles of needles that were aligned in parallel to form a single layer used for the measurements. A saturation Pulse (10000 μmol photons $m^{-2} s^{-1}$ for 300 ms) was applied with a sequence of increasing actinic light intensity from 0 to 2000 μmol photons $m^{-2} s^{-1}$ with 30 s intervals. Each measurement was made with four-to-six replicates. Photosynthetic parameters were calculated as described in refs. [48,49]. Y(ND), quantum yield of non-photochemical energy dissipation in PSI reaction centers that are limited due to a shortage of electrons (donor-side limitation). Y(NA), quantum yield of non-photochemical energy dissipation in PSI reaction center s that are limited due to shortage of electron acceptors (acceptor-side limitation). Relative yield of AEF (Y(AEF)), representing the Δ flow in conifers between PSI and PSII mainly contributed by CET and pseudo-CET[29]. Y(AEF) was calculated as Y(AEF) = Y(I) – Y(II).

For measurements mimicking fluctuating light, needles from mature *P. sylvestris* and *P. abies* trees were collected on 9th April 2018. The needles from *P. sylvestris* and *P. abies* seedlings grown under 22 °C were used as controls. The chlorophyll a and P700 signal were monitored after 2 min dark, followed by four cycles of 5 min low light (58 μmol photons $m^{-2} s^{-1}$) and 1 min high light (1599 μmol photons $m^{-2} s^{-1}$). For the measurements under the steady-state conditions, the light intensity was set for 5 min either at high light (1599 μmol photons $m^{-2} s^{-1}$) or moderate light (536 μmol photons $m^{-2} s^{-1}$). Each measurement was made with four-to-six replicates. To address the effect of inhibition of PGR5-dependent cyclic electron transport, branches from *P. sylvestris* and *P. abies* were collected on 10th of April 2019. Detached needles were soaked in distilled water or water containing 200 μM antimycin A (AA). Needles were vacuum infiltrated for 15 min, and the treatment repeated four times. Needles were sandwiched with wet tissue paper and incubated in the dark for 30 min before applying fluctuating light measurements. After the initial measurements, branches were allowed to recover in room temperature for 24 h, and measured again. Each measurement was made with three replicates.

**Gas-exchange analysis**. The net $CO_2$ assimilation rate ($A_N$) was measured with the gas-exchange system (Li-6400xt, Li-Cor, Lincoln, NE, USA). $A_N$ over the seasons was measured at a $CO_2$ concentration ($C_a$) of 800 μmol mol$^{-1}$ and a photon flux density of 1200 μmol photons $m^{-2} s^{-1}$ with branches collected from the field in the morning. The cuvette temperature was set to 23 °C, and the airflow was set to 400 μmol s$^{-1}$. $A_N$ versus the calculated intercellular $CO_2$ partial pressure ($A/C_i$ curve) was measured with samples collected on 19th April 2018 and 8th April 2019. The assimilation rate was assessed after 2 to 3 min of exposure to $CO_2$ concentrations of 400, 200, 150, 100, 50, 400, 650, 800, 1000, and 1200 μmol mol$^{-1}$ $CO_2$, based on a protocol described by Chang et al.[50]. All measurements were performed at 25 °C and 1400 μmol photons $m^{-2} s^{-1}$ with four-to-six biological replicates. The same set of samples was allowed to recover in room temperature for 24 h after the initial measurements, and then measured again with the same protocol. Cold-acclimated seedlings were transferred either to warm (22 °C, indicated in black) or cold (5 °C, indicated in gray) chambers for 4 weeks, and $A/C_i$ curves were determined again. The maximum carboxylation rate ($V_{cmax}$) and maximum electron transport rate ($J_{max}$) were estimated according to Sharkey et al.[51].

**Transmission electron micrographs**. Samples were prepared with a modified procedure according to Jonsson et al.[52]. In all, 0.5 mm-long, cross-sectional needle samples were cut from the middle region of five needles and placed into tubes containing a fixation solution (2.5% glutaraldehyde, 4% paraformaldehyde in 0.1 M cacodylate buffer, pH 7.2–7.4) and kept at 4 °C overnight. The samples were then rinsed two times in 0.1 M cacodylate buffer (pH 7.2–7.4) for 10 min and fixed in 1% $OsO_4$ dissolved in the cacodylate buffer for 2 h in darkness. After

wash with MQ water 2 × 10 min, tissue samples were dehydrated in graded ethanol series (50%, 60%, 70%, 80%, 90%, 100%) followed by propylene oxide fixation for 20 min, and then embedded in Spurr epoxy resin medium. After trimming, sections for three samples per species for imaging were cut with an EM UCF 7 Ultra Microtome (Leica) using diamond knife, then mounted on copper grids and stained with 5% uranyl acetate (dissolved in MQ water) and Reynolds lead citrate. Whole cells and chloroplasts were photographed with a digital camera (Gatan Orius CCD and Ceta CMOS) connected to a transmission electron microscope (JEOL 1230 and FEI Talos L 120 C). The digital images were analyzed using ImageJ (version 1.51j8, National Institute of Health) and Photoshop CC (version 2017.0.1, Adobe software). The number and total area of plastoglobules per chloroplast from the transmission electron micrographs ($n = 8$–$12$) were quantified from three independent experiments using the program ImageJ software[53].

**Carbohydrates and lipid peroxidation analysis**. Soluble sugars including sucrose, glucose, and fructose were determined in ethanol extracts as described by Stitt et al.[54]. The pellets of the ethanol extraction were used for starch determination with methods described by Smith and Zeeman[55] with slight modification. The incubation time for starch degradation was increased from 4 h to 12 h. The level of general lipid peroxidation was measured using the modified thiobarbituric acid-malondialdehyde (TBA-MDA) method[56]. Needles were powdered in liquid nitrogen and homogenized in 5% TCA. The homogenate was centrifuged at 12,000 g for 15 min. Reaction buffer of 300 μl 0.65% (w/v) thiobarbituric acid (TBA) containing 20% TCA and 0.9 μl 3.3% (w/v) butylated hydroxytoluene was added to 300 μl aliquots of supernatant. The mixture was heated at 95 °C for 30 min, and then rapidly cooled in an ice-bath. Absorption was measured at 532 , 600 , and 440 nm with three replicates, respectively. The concentration of malondialdehyde was calculated as per fresh weight.

**Protein separation and immunoblotting**. Proteins were extracted as described by Wang et al.[57]. One gram of needles were pulverized to a fine powder using mortar and pestle under liquid nitrogen together with 0.05 g polyvinylpolypyrrolidone. In all, 0.1 g tissue powder was washed with 2 ml 10% trichloroacetic acid (TCA)/acetone buffer, followed by a methanol buffer (80% methanol with 0.1 M ammonium acetate) wash and an 80% acetone wash step. Subsequently, total of 1.8 ml mixture of 1:1 phenol (pH 8.0)/SDS buffer (30% sucrose, 2% SDS, 0.1 M Tris-HCl, pH 8.0, 5% 2-mercaptoethanol) were added to the tube to extract proteins from dry pellets. After centrifuging at 16,000 g, 4 °C for 3 min, 0.4 ml phenol phase extraction were mixed with 1.6 ml methanol buffer and then centrifuged at 16,000 g, 4 °C for 3 min. The pellets were washed twice with methanol and 80% acetone. Final protein pellets were dissolved in 0.2 ml Laemmli sample buffer. Total proteins were quantified with a Pierce BCA protein assay kit (Thermo Scientific). SDS-PAGE was performed in 12% polyacrylamide gels with 25 μg total protein loaded per well. The gels were stained with Coomassie Brilliant Blue R as a loading control. Proteins were transferred to a polyvinylidene difluoride (PVDF) membrane (GE). Immunoblot analysis were performed with antibodies raised against the PGR5 (dilution 1:1000) and PGRL1 (dilution 1:1000) of *Arabidopsis* (Agrisera) and antibody of FLVB (dilution 1:2000) which was provided by Dr. Shikanai[19]. Secondary antibody was anti-rabbit (Agrisera). The protein sequences of PGR5 and PGRL1 are highly conserved between Scots pine and Norway spruce (Supplementary Fig. 11), and the peptide targets for PGR5 antibody in these two species were identical. Each analysis was repeated with three biological replicates. Protein levels were quantified from three independent experiments using the program ImageJ software[53].

**Gene expression analysis**. Needles from mature *P. sylvestris* and *P. abies* trees were collected during February 2017 to May 2017. The total RNA was isolated with the Spectrum$^{TM}$ Plant Total RNA Kit (Sigma Aldrich) following the protocols of the manufacturer. RNA was quantified using the NanoDrop spectrophotometer (Thermo Scientific). cDNA synthesis and real-time PCR performed as described by Díaz et al.[58]. Gene-specific primers were designed with Primer3[59] and listed in Supplementary Table 1. Three biological and three technical replicates were performed for each experiment. Data analysis was performed with CFX manager software (Bio-Rad). Relative expression values were normalized against the reference gene PP2A (locus name: lcl|PgdbPsylvestris_72087 for *P. sylvestris* and lcl|MA_10426823g0010 for *P. abies*). All values were related to the February samples within the same species.

**Reporting summary**. Further information on research design is available in the Nature Research Reporting Summary linked to this article.

## Data availability
The underlying Figs. 1, 2b–d, 3, 4 and 5 and Supplementary Figs. 2, 3, 4a, 5, 7, 8, 9 and 10 are provided as a Source Data file. Any other data that support the findings of this study are available within the paper and its supplementary files or are available from the corresponding author(s) upon request.

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

## Acknowledgements

Dr. Shikanai is gratefully acknowledged for the kind gift of the antibodies against FLV and PGR5. Dr. Daria Chrobok (https://www.dariasciart.com/) is acknowledged for the artwork in Fig. 6. The authors acknowledge the facilities and technical assistance of the Umeå Core Facility Electron Microscopy (UCEM) at the Chemical Biological Centre (KBC), Umeå University a part of the National Microscopy Infrastructure NMI. Funding from the JCK foundation JCK-1610 (QY) and the project "TC4F—Trees and Crops for the Future" funded through the Swedish government's Strategic Research Environment "Sustainable use of Natural Resources" is acknowledged. Open access funding provided by Umeå University.

## Author contributions

Å.S., Q.Y., N.E.B. and V.H. designed the research. Q.Y., N.E.B., C.H.-C. and N.L. performed the research. All authors contributed to data analysis, writing of the paper, and reviewed and approved the final version of the paper.

## Competing interests

The authors declare no competing interests.

## Additional information

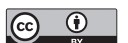

