## [Peer Review File · Nature Communications]

Reviewers' comments:

Reviewer #1 (Remarks to the Author):

Little is known how boreal conifers protect their photosynthetic electron transport chain at low temperatures against photooxidative damage. Since boreal forests represent 22% of global CO₂ storage, it is an important issue to study acclimation processes in these trees. Scots pine and Norway spruce were chosen for this study and differences were observed for alternative electron transport activities. In contrast to spruce, pine activated alternative electron flow in the critical period of late winter to early spring to protect the photosynthetic apparatus while spruce suffered more from photooxidative damage. Electron microscopy showed differences in the organization of the thylakoid membrane and the number of plastoglobuli in early spring needles between the two species. Photosynthesis and electron transport were characterized by chlorophyll fluorescence, P700 measurements and gas exchange. Gas exchange was studied both, in needles with down-regulated photosynthetic electron transport and in needles that were warmed and showed activation of photosynthetic electron transport. A clear difference in assimilation between the two species is shown, indicating down regulation of photosynthetic electron transport in pine and not in spruce. There is clearly a difference between pine and spruce in photosystem I. It is clearly shown that pine suffers less from PSI acceptor side limitation and photooxidative damage. It is very interesting that the amount of PGR5 is much lower in spruce than in pine. However, it is not convincingly shown that this is due to cyclic electron flow.

A convincing determination of cyclic electron flow is difficult. It is heavily debated if the ratio between Y(I) and Y(II) can be interpreted as a sign of cyclic electron flow. One has to keep in mind that Y(I) is overestimated with the Dual-PAM. Additional measurements like electrochromic shift are needed to show that it is indeed cyclic electron flow that is responsible for the observed differences. To measure only Y(I) and Y(II) is insufficient. It may well be that something else is changed in the properties of PSI and that, for example, charge recombination reactions within PSI are increased in pine and not in spruce. Or that Mehler reaction/activity of flavodiiron proteins is higher in pine than in spruce. This would give the same effect on the measured parameters. The effect of antimycin is not very convincing. The decrease in Y(I) by antimycin is rather small and even smaller in pine than in spruce. The effect of the recovery from antimycin treatment is hard to see (Fig. 5d). The signals in spruce are much larger than for pine (and therefore easier to see) and it seems that only a part of the signal is recovered after 24h. At least for the early time point (2-7 min), the effect seems to be similar in spruce and pine. It is an interesting observation that the amount of PGR5 is lowered after room temperature recovery (Fig. 5 e).

Immunoblots (Fig. 5a, b): The question arises why PGR5 levels are still very high in June. Fig. 5a: It would be nice to show data also on PGRL1 to exclude that the antibodies directed against PGR5 don't recognize well the spruce protein. In addition, the level of the NDH complex should be checked. It may also differ between the two species and depending on the time of the year.

Minor comments

I. 168, the key paper by Sonoike (a FEBS letters publication showing the loss of FeS clusters by EPR= should be cited in addition to ref 24.

L. 193, 194. In Fig. 5d, the Y(I) in pine is 0.6 and in spruce 0.8. How to explain this compared with Fig. 4d?

Reviewer #2 (Remarks to the Author):

The authors investigated PGR5-related cyclic electron flow in Scots pine and Norway spruce, regarding photooxidative damage from winter to spring. The topic on springtime photodamage is very relevant for understanding the nature of evergreen conifer species, since they should suffer

some imbalance between recovery from winter dormancy and photosynthetic reactivation during springtime. The finding that the two conifer species have contrasting mechanisms to manage spring recovery of photosynthesis is novel. The reviewer has just a few comments to be addressed as follows.

1. The authors well discussed the mechanism of Norway spruce coping with springtime photodamage, i.e. alternative electron flow (AEF) under down-regulated photosynthetic capacity. The reviewer recommends the authors to discuss the strategy of Norway spruce during the critical spring recovery phase as well, from the view point of forest succession. As the authors describe, Scots pine and Norway spruce is classified as pioneer and late successional climax species, respectively. As late successional species is often found in the forest understory, photosynthetic carbon gain during early spring before the flush of canopy trees (especially in the case of deciduous tree species such as European Beech) is substantially important for growth and survival of the forest-floor seedlings. In this context, relatively higher An in Norway spruce observed early in the spring (Fig. 1a, b, e) may be a strategy of this species for efficient carbon gain. Furthermore, shade-tolerance or shade-preference of this species may be a way of circumventing photooxidative damage without enhanced AEF.

2. Budbreak, i.e. the onset of new shoot development, is physiologically relevant for evergreen conifer trees, since pre-existing shoots act as a sink of photosynthate before budbreak, but as a source of new shoot development after budbreak; starch content changes drastically in both species around budbreak (Egger et al., 1996; Wyka et al., 2016). Therefore, the authors should indicate the date of budbreak for each species in the related figures.

Minor comments

1. [Line 320] Light intensity ($150 \mu\text{mol m}^{-2} \text{s}^{-1}$) for the growth chamber seedlings was substantially low ($\approx 10\%$ of full sunlight). In this case, it should be noted that needles from mature trees were sun-acclimated, but those from growth chamber seedlings were shade-acclimated.

2. [Line 333] Y(NA) and Y(ND) should be explained here.

3. [Line 352] Please explain why using Ca of $800 \mu\text{mol mol}^{-1}$, twice higher than the ambient air.

4. [Figure 6] Based on Figure 1c, no significant difference seems to be observed in ETR(II) of Scots pine between spring and summer. Please check the blue lines through PSII.

References

Egger B, Einig W, Schlereth A, Wallenda T, Magel E, Loewe A, Hampp R (1996) Carbohydrate metabolism in one- and two-year-old spruce needles, and stem carbohydrates from three months before until three months after bud break. *Physiologia Plantarum*, 96, 91–100.

Wyka TP, Żytkowiak R, Oleksyn J (2016) Seasonal dynamics of nitrogen level and gas exchange in different cohorts of Scots pine needles: a conflict between nitrogen mobilization and photosynthesis? *European Journal of Forest Research*, 135, 483–493.

Reviewer #3 (Remarks to the Author):

Overall, I found that the work is novel both for how these co-occurring species deal with seasonal changes in climate, but also for its broad implications for how these two globally common species respond to changes in temperature. The data are comprehensive, spanning anatomy, fluorescence and gas exchange, biochemistry and gene expression results from a suite of interconnected field

and chamber experiments. The paper is well written and the arguments are clear and convincing.

Having said that, I do have some comments that would clarify the analyses. The statistical approach is quite mixed - some data have no stats, which is unacceptable. While many of the results are obviously significant, all of them need to be backed with stats.

Figure 3 - in panel b, there's a marked and unusual disjunct in the A_{Ci} data around 250 ppm C_i. Why is this? If it's a break caused by the measurements protocol (i.e from moving down in CO₂ and then moving back up to high CO₂), then I would be concerned about the quality of the data. If it's a real biological phenomenon, then I think this needs some explanation. The matching data in panel c does not have this issue, which gives me confidence that it's more likely biological than protocol-based.

The authors discuss some of the differences in the ecology of these species. Are there other data pointing to differences in how they might cope with environmental conditions? Is the "superior" strategy of pine indicative of a greater capacity to cope with environmental change in general, or is its advantage limited to seasonal responses of photosynthesis?

While I think that the novel contrasting strategies outlined here are important, photosynthetic recovery and performance are not growth. As such, the legend in Figure 6 need to be toned down.

Minor comments

Figure 2 - there are no starch granules in the figure that I can see, so no need for the abbrev in the legend.

Figure 4 - relabeling the figure legend inside the figure might help keep readers clear. Saying Pine vs Pine - control is less helpful when there are so many experiments with so many combinations of "control". Perhaps like "Pine - field, April" and "Pine - chamber controls"? Similar tweaks in the other figures would help readers see which data are in which figures.

Figure 6 - here, and elsewhere, please define all abbreviations and symbols used. I can read the figure only because I know the processes being illustrated.

Response to Reviewer 1

It is very interesting that the amount of PGR5 is much lower in spruce than in pine. However, it is not convincingly shown that this is due to cyclic electron flow. A convincing determination of cyclic electron flow is difficult. It is heavily debated if the ratio between Y(I) and Y(II) can be interpreted as a sign of cyclic electron flow. One has to keep in mind that Y(I) is overestimated with the Dual-PAM. Additional measurements like electrochromic shift are needed to show that it is indeed cyclic electron flow that is responsible for the observed differences. To measure only Y(I) and Y(II) is insufficient. It may well be that something else is changed in the properties of PSI and that, for example, charge recombination reactions within PSI are increased in pine and not in spruce. Or that Mehler reaction/activity of flavodiiron proteins is higher in pine than in spruce. This would give the same effect on the measured parameters. The effect of antimycin is not very convincing. The decrease in Y(I) by antimycin is rather small and even smaller in pine than in spruce. The effect of the recovery from antimycin treatment is hard to see (Fig. 5d). The signals in spruce are much larger than for pine (and therefore easier to see) and it seems that only a part of the signal is recovered after 24h. At least for the early time point (2-7 min), the effect seems to be similar in spruce and pine. It is an interesting observation that the amount of PGR5 is lowered after room temperature recovery (Fig. 5 e).

- Thank you for these insightful comments. We have addressed them to the best of our abilities. We must note that the main message of the work presented is how differential regulation of PSI function and CO₂ assimilation capacity underpin the photosynthetic recovery process (winter to spring and summer) in spruce and pine. Our intention was not to present evidence for the molecular bases of cyclic electron transport (CET). We might have mis-lead the reader to get this impression by overstating some of the differences between pine and spruce and we sincerely apologize for that. We have now carefully gone through the manuscript to correct for such phrasings in the text.

Specifically, our findings indicate that the regulation of PGR5 expression in pine is one of the central aspects that provides this species with a better protection against PSI acceptor-side limitation compared to spruce. The other aspect that contributes to this capacity is the presence of flavodiiron proteins (FLVs) which also provides a large electron sink at the PSI acceptor-side. However, FLV levels remain stable during the whole period of the study according to our data (Fig. 5a and 5b). The coordinate function of PGR5 and FLV is the mechanism behind the dynamic capacity observed in pine to avoid the redox imbalance as shown by the fluctuating light experiment (Fig. 4). The importance of these two components has been widely demonstrated (Allahverdiyeva et al., 2013; Yamamoto et al., 2016; Shimakawa et al., 2017). The parameter, alternative electron flow (AEF), is dependent on both PGR5 and FLV-mediated non-photosynthetic electron flows.

Although the induction of PGR5 strongly influences AEF during winter to spring transition, we cannot attribute the better performance of pine to cyclic electron transport (CET) - as you correctly pointed out, the molecular bases of CET still remains elusive. In addition, as you acknowledged, there is no efficient method to quantify CET, not even by electrochromic shift (ECS) in the presence of FLVs. The measurement of *pmf* and their components ΔpH and $\Delta\Psi$ cannot provide reliable support for CET. Yamamoto and coworkers have used ECS to evaluate the contribution of FLV in the *pgr5* and wild-type backgrounds and concluded that both PGR5 and FLV contribute to *pmf* (Yamamoto et al., 2016; Shikanai and Yamamoto, 2017). We did not claim that “ratio between Y(I) and Y(II) can be interpreted as a sign of cyclic electron flow”, rather as an indicator of AEF to photosynthetic LEF. Recently, Grebe and co-workers (Grebe et al., 2019) revealed the existence of a large subpopulation of PSI, named PSI*, with different antenna compositions. The authors proposed that this PSI* subpopulation can generate a photoprotective mechanism including FLV and/or as a supplementary structure under demanding conditions. The result presented by Grebe et al is in line with our view that the ratio $Y(AEF) = Y(I)-Y(II)$ is not a specific indicator of CEF, but an indicator of alternative electron fates at the PSI acceptor-side. Although other variables (e.g. specific PSI structures, different FLV-PSI affinity constants or species-specific variations in Mehler reaction) might contribute to the differences in PSI function between pine and spruce, the combined effect of PGR5 and FLV on the PSI redox status can be considered as major contributors to the photoprotective mechanisms during the winter-spring transition. The Antimycin A treatment was conducted as a complementary approach to investigate PSI function during the spring to summer transition. Antimycin A (AA) has been widely used as inhibitor of PGR5 function (Labs et al., 2016; Takagi et al., 2018). In combination, the change in PGR5 protein abundance and the effect of AA treatment supports the link between PGR5 activity and Y(I).

References: Allahverdiyeva, Y., H. Mustila, M. Ermakova, L. Bersanini, P. Richaud, G. Ajlani, N. Battchikova, L. Cournac and E. M. Aro (2013). "Flavodiiron proteins Flv1 and Flv3 enable cyanobacterial growth and photosynthesis under fluctuating light." *Proc Natl Acad Sci U S A* **110**(10): 4111-4116. Grebe, S., A. Trotta, A. A. Bajwa, M. Suorsa, P. J. Gollan, S. Jansson, M. Tikkanen and E. M. Aro (2019). "The unique photosynthetic apparatus of Pinaceae: analysis of photosynthetic complexes in *Picea abies*." *J Exp Bot* **70**(12): 3211-3225. Labs M, Ruhle T, Leister D (2016). "The antimycin A-sensitive pathway of cyclic electron flow: from 1963 to 2015." *Photosynth Res* **129**: 231-238. Shimakawa, G., K. Ishizaki, S. Tsukamoto, M. Tanaka, T. Sejima and C. Miyake (2017). "The Liverwort, *Marchantia*, Drives Alternative Electron Flow Using a Flavodiiron Protein to Protect PSI." *Plant Physiol* **173**(3): 1636-1647. Shikanai, T. and H. Yamamoto (2017). "Contribution of Cyclic and Pseudo-cyclic Electron Transport to the Formation of Proton Motive Force in Chloroplasts." *Mol Plant* **10**(1): 20-29. Takagi, D., K. Ifuku, T. Nishimura and C. Miyake (2018). "Antimycin A inhibits cytochrome b559-mediated cyclic electron flow within photosystem II." *Photosynth Res* **139**(1-3): 487-498. Yamamoto, H., S. Takahashi, M. R. Badger and T. Shikanai (2016). "Artificial remodelling of alternative electron flow by flavodiiron proteins in *Arabidopsis*." *Nat Plants* **2**: 16012.

Immunoblots (Fig. 5a, b): The question arises why PGR5 levels are still very high in June. Fig. 5a: It would be nice to show data also on PGRL1 to exclude that the antibodies directed against PGR5 don't recognize well the spruce protein. In addition, the level of the NDH complex should be checked. It may also differ between the two species and depending on the time of the year.

Why PGR5 levels are still very high in June: In the publication by Suorsa et al from 2012 it was shown that high light and fluctuating light could induce the accumulation of the PGR5 protein, which was also suggested to indicate a protective function of PGR5 under such stress conditions. In Northern Sweden, the days are very long in June (about 20 hours light per day) and the temperature could still occasionally be low also at this time of year, this could potentially generate conditions to induce the accumulation of PGR5 resulting in maintaining rather high levels of the protein also after the critical spring period.

Antibodies directed against PGR5 don't recognize well the spruce protein: The PGR5 sequence alignment between pine and spruce as was shown in the previous Figure S9 (now Figure S11) demonstrated that PGR5 from the two conifer species are highly conserved. They are also both very similar to the Arabidopsis PGR5. The same is true for PGRL1, and the PGRL1 sequence alignment for pine and spruce has now been included in the new Fig. S11.

It would be nice to show data also on PGRL1: Thank you for the good suggestion to investigate also the levels of the PGRL1 protein during the spring summer transition. Figure 5 has been revised to now also include the PGRL1 Western blot. The result for the PGRL1 is very similar to what was shown for PGR5. The levels of PGRL1 accumulate strongly during the critical spring period to then decline in the summer months. Similar to PGR5, pine demonstrated higher levels of PGRL1 compared to spruce. The new data is presented in the new Figure 5a and 5b.

The level of the NDH complex should be checked: The literature presents strong evidence that the NDH complex is absent in conifers. It was reported that all chloroplast encoded ndh genes are lost in the pine family (Braukmann et al., 2009). We also further investigated the ndh genes in the chloroplast genomes of Scots pine and Norway spruce (NCBI Reference Sequence: NC_035069.1 and NC_021456.1) and all plastid encoded ndh genes were either pseudogenes or lost. In addition, Grebe et al, 2019 demonstrated that no PSI-NDHmc complex could be detected in thylakoid protein preparations from Picea abies (Fig. 5 in Grebe et al, 2019).

References: Suorsa, M., S. Jarvi, M. Grieco, M. and et al. (2012). "PROTON GRADIENT REGULATION5 is essential for proper acclimation of Arabidopsis photosystem I to naturally and artificially fluctuating light conditions." Plant Cell **24(7): 2934-2948. Braukmann, T. W., M. Kuzmina and S. Stefanovic (2009). "Loss of all plastid ndh genes in Gnetales and conifers: extent and evolutionary significance for the seed plant phylogeny." Curr Genet **55**(3): 323-337. Grebe, S., A. Trotta, A. A. Bajwa,**

M. Suorsa, P. J. Gollan, S. Jansson, M. Tikkanen and E. M. Aro (2019). "The unique photosynthetic apparatus of Pinaceae: analysis of photosynthetic complexes in *Picea abies*." *J Exp Bot* **70**(12): 3211-3225.

Minor comments:

L. 168, the key paper by Sonoike (a FEBS letters publication showing the loss of FeS clusters by EPR= should be cited in addition to ref 24.

- We apologize for overlooking the paper by Sonoike et al. 1995. This publication and the work presented therein pioneered the studies of PSI photoinhibition. The publication is now included in our references.

L. 193, 194. In Fig. 5d, the Y(I) in pine is 0.6 and in spruce 0.8. How to explain this compared with Fig. 4d?

- The samples in Fig. 4d and Fig. 5d were collected in 2017 and 2018, respectively. In the spring 2018 the recovery of photosynthesis in Norway spruce was observed slightly earlier compared to Scots pine. All samples were collected in April, but due to marginally warmer temperatures during this period (which is also seen in the earlier bud break 2018 compared to 2017) slightly higher Y(I) was observed in spruce 2018 compared to 2017. The difference between the years for Y(I) was only observed in Norway spruce under low light and we feel that the broad reproducibility of these findings from field collected samples across two different growing seasons strengthens the findings we report.

Response to Reviewer 2

1. The authors well discussed the mechanism of Norway spruce coping with springtime photodamage, i.e. alternative electron flow (AEF) under down-regulated photosynthetic capacity. The reviewer recommends the authors to discuss the strategy of Norway spruce during the critical spring recovery phase as well, from the view point of forest succession. As the authors describe, Scots pine and Norway spruce is classified as pioneer and late successional climax species, respectively. As late successional species is often found in the forest understory, photosynthetic carbon gain during early spring before the flush of canopy trees (especially in the case of deciduous tree species such as European Beech) is substantially important for growth and survival of the forest-floor seedlings. In this context, relatively higher An in Norway spruce observed early in the spring (Fig. 1a, b, e) may be a strategy of this species for efficient carbon gain. Furthermore, shade-tolerance or shade-preference of this species may be a way of circumventing photooxidative damage without enhanced AEF.

- Thank you for this very interesting comment. We have included a section in the discussion to bring up these important points. The new text is high-lighted in the document and can be found in the last paragraph of the discussion.

2. Budbreak, i.e. the onset of new shoot development, is physiologically relevant for evergreen conifer trees, since pre-existing shoots act as a sink of photosynthate before

budbreak, but as a source of new shoot development after budbreak; starch content changes drastically in both species around budbreak (Egger et al., 1996; Wyka et al., 2016). Therefore, the authors should indicate the date of budbreak for each species in the related figures.

- Thank you for this suggestion, we have included data for budbreak for both species from the experimental site in Figure 1b and 3a, and supplementary Figure S2. For spruce budburst was June 15 (2017) and May 28 (2018) (average Krutzsch index 3, which represents the budburst stage). For pine it was June 8 (2017) and May 22 (2018) (average apical shoot length reached 20 mm).

Minor comments

1. [Line 320] Light intensity ($150 \mu\text{mol m}^{-2} \text{s}^{-1}$) for the growth chamber seedlings was substantially low ($\approx 10\%$ of full sunlight). In this case, it should be noted that needles from mature trees were sun-acclimated, but those from growth chamber seedlings were shade-acclimated.

- We included “sun-acclimated needles” in the legend of figure 4 and main text. We would prefer to not refer to the plants from the climate chamber as “shade-acclimated” because (i), the light intensity of $150 \mu\text{mol m}^{-2} \text{s}^{-1}$ is high enough to support fully active photosynthesis in conifers (Verhoeven, 2016); (ii) the yield of PSII under PAR of $131 \mu\text{mol m}^{-2} \text{s}^{-1}$ showed less than 40% of maximum photochemical efficiency left in summer samples of pine and spruce by light curve measurement indicating that the light intensity is close to saturated light for pine and spruce; (iii) when we use “shade-acclimated” to describe plant growth conditions, it would imply changes to the light quality, far-red to red light ratio, and this was not the case in our experiments.

Reference: Verhoeven, A. S., A. Kertho and M. Nguyen (2016). "Characterization of light-dependent regulation of state transitions in gymnosperms." *Tree Physiol* **36**(3): 325-334.

2. [Line 333] $Y(NA)$ and $Y(ND)$ should be explained here.

-Thank you, we corrected this.

3. [Line 352] Please explain why using Ca of $800 \mu\text{mol mol}^{-1}$, twice higher than the ambient air.

- We used $800 \mu\text{mol mol}^{-1}$ to assess the full capacity in the needles for CO_2 assimilation during the spring and summer period as we were interested in potential differences in the enzymatic capacities in the Calvin cycle etc. between the two species during the recovery phase. Our data showed the CO_2 assimilation rate was very low (below zero) in the early spring samples (March and April) under $400 \mu\text{mol mol}^{-1}$. However, we have now included the data collected for $400 \mu\text{mol mol}^{-1}$ as the new supplementary Figure S2 and refer to it in the text.

4. [Figure 6] Based on Figure 1c, no significant difference seems to be observed in $ETR(II)$ of Scots pine between spring and summer. Please check the blue lines through PSII.

- According to the light response curve presented in the new Supplementary Figure S10, the ETR(II) in pine in summer samples was higher compared to the spring samples under different light intensities, especially from 100 to 500 $\mu\text{mol m}^{-2} \text{s}^{-1}$ the differences were significant ($p < 0.05$). In Figure 1c, the ETR(II) was calculated under saturated light intensity of 1292 $\mu\text{mol m}^{-2} \text{s}^{-1}$. Due to the large variation of ETR(II) under high light, the differences became less pronounced as presented in Figure 1c.

Response to Reviewer 3

The statistical approach is quite mixed - some data have no stats, which is unacceptable. While many of the results are obviously significant, all of them need to be backed with stats.

- Thank you for pointing this out. It was a mistake on our part. We have now included proper and uniform statistical analysis using ANOVA for Figure 2, 3, 4 and 5, and also supplementary Figure S3 and S7.

Figure 3 - in panel b, there's a marked and unusual disjunct in the ACi data around 250 ppm Ci. Why is this? If it's a break caused by the measurements protocol (i.e from moving down in CO2 and then moving back up to high CO2), then I would be concerned about the quality of the data. If it's a real biological phenomenon, then I think this needs some explanation. The matching data in panel c does not have this issue, which gives me confidence that it's more likely biological than protocol-based.

- Thank you for this comment. We should have made a note about this in the manuscript. The observed gap in Figure 3b is biological. The needles were collected directly from the field for Figure 3b. Stomata conductance values indicated a greater sensitivity of stomata to cold, both in pine and spruce. The revised table included in the new version of Figure 3e shows the stomatal conductance values (g_s), which were 0.01 and 0.03 $\text{mol CO}_2 \text{m}^{-2} \text{s}^{-1}$ for Pine and Spruce field samples, respectively, and 0.06 $\text{mol CO}_2 \text{m}^{-2} \text{s}^{-1}$ for both recovered species. The response to the warm temperature was indeed slow and would explain the observed a gap during the measurement.

Moreover, at low concentrations of CO_2 , most A_N values are near zero. This suggests that Rubisco, Calvin and Krebs cycles, and diffusion (aquaporins) etc., needs to be activated in response to the warm temperatures. Before each measurement, we placed the needles in the leaf chamber of the IRGA under 1400 $\mu\text{mol m}^{-2} \text{s}^{-1}$ light intensity and 25 °C to equilibrate for 15 minutes. The field samples need longer time to pre-accumulate to the conditions used for the measurements. However, we also wanted to limit the recovery of photosynthesis to reflect the photosynthetic abilities in the field. We have included a note in the manuscript about the disjunct in the ACi data referring to the g_s , line 168 and onwards.

The authors discuss some of the differences in the ecology of these species. Are there other data pointing to differences in how they might cope with environmental conditions? Is the "superior" strategy of pine indicative of a greater capacity to cope with environmental change in general, or is its advantage limited to seasonal responses of photosynthesis?

- It was recently shown that in response to increased seasonal temperatures and elevated CO₂ Scots pine is more able compared to Norway spruce to acclimate photosynthesis and respiration. Pine demonstrated increasing growth in response to temperature increases as high as +8 °C, whereas spruce showed minimal capacity to acclimate energy metabolism and suffered growth losses at elevated seasonal temperatures (Kurepin et al, 2018). We feel that pine generally is the more responsive of the two species and better able to acclimate to environmental fluctuations and we have included this information in the discussion. The new text is high-lighted in the word file and found in the last paragraph of the discussion.

Reference: Kurepin, L.V. Stangl, Z.R., Ivanov, A.G., Bui, V., Marin, M., Huner, N.P.A., Öquist, G., Way, D., Hurry, V. (2018) Contrasting acclimation abilities of two dominant boreal conifers to elevated CO₂ and temperature. *Plant Cell and Environment*. **41**: 1331-1345. DOI:10.1111/pce.13158

While I think that the novel contrasting strategies outlined here are important, photosynthetic recovery and performance are not growth. As such, the legend in Figure 6 need to be toned down.

- Thank you, you are correct. We have re-written the legend of Figure 6 to focus primarily on photosynthesis.

Minor comments

Figure 2 - there are no starch granules in the figure that I can see, so no need for the abbrev in the legend.

- Thank you, this was a mistake and we have changed the figure legend.

Figure 4 - relabeling the figure legend inside the figure might help keep readers clear. Saying Pine vs Pine - control is less helpful when there are so many experiments with so many combinations of "control". Perhaps like "Pine - field, April" and "Pine - chamber controls"? Similar tweaks in the other figures would help readers see which data are in which figures.

- We apologize for the messy labeling, we have now changed the labeling in the new revised Figure 4.

Figure 6 - here, and elsewhere, please define all abbreviations and symbols used. I can read the figure only because I know the processes being illustrated.

- We apologize for this and have now included a list of all the abbreviation used in the manuscript.

REVIEWERS' COMMENTS:

Reviewer #1 (Remarks to the Author):

The response to my comments and the revised submission have adequately addressed my concerns and questions. I recommend to accept the manuscript for publication.

I just want to add that I had made a mistake in my first review. $Y(I)$ is underestimated by the P700 measurements and not overestimated thanks to acceptor side limitation upon onset of illumination.

Anja Krieger-Liszkay

Reviewer #2 (Remarks to the Author):

The authors have adequately addressed all of my comments. The reviewer recommends the revised manuscript for publication.

Reviewer #3 (Remarks to the Author):

Overall, I find that the revisions have dealt with most of the comments raised by myself and the other reviewers. I think the work is novel and of potentially great impact given the importance of the species involved and their role in the global carbon cycle.

Having said that, I have a few comments:

1. The title should read "Two dominant..." not "The dominant...". These species are the dominant Scandinavian boreal conifers, but not necessarily the most common global boreal conifers. Ditto for line 279.

Line 83 - please define PQ.

Figure 1 - I would argue that needles from a single tree are not independent replicates. I suggest rephrasing.

Figure 3 - low g_s will not generate a break in the A_{Ci} curve, it will just shift the C_i values down for a given C_a . So that text can be deleted from the paper (line 167 and on). Instead, the break appears to be where the measurements went from decreasing C_i below 400 ppm to raising it above that C_a level. That instead indicates that the continued exposure to warmer temperatures stimulated photosynthetic processes over the time that the curves were made. Provided that the V_{cmax} data are generated from only low C_i data and the J_{max} data from high C_i data, this may be OK. But there should still be more data points at low C_i - are data points missing?

Response to Reviewer #1:

The response to my comments and the revised submission have adequately addressed my concerns and questions. I recommend to accept the manuscript for publication.

I just want to add that I had made a mistake in my first review. $Y(I)$ is underestimated by the P700 measurements and not overestimated thanks to acceptor side limitation upon onset of illumination.

Thank you for your explanation.

Response to Reviewer #2:

The authors have adequately addressed all of my comments. The reviewer recommends the revised manuscript for publication.

Thank you very much.

Response to Reviewer #3:

1. The title should read "Two dominant..." not "The dominant...". These species are the dominant Scandinavian boreal conifers, but not necessarily the most common global boreal conifers. Ditto for line 279.

Thank you for your suggestion, we have changed it.

2. Line 83 - please define PQ.

The definition of PQ was added.

3. Figure 1 - I would argue that needles from a single tree are not independent replicates. I suggest rephrasing.

Thank you, we have rephrased this.

4. Figure 3 - low g_s will not generate a break in the AC_i curve, it will just shift the C_i values down for a given C_a . So that text can be deleted from the paper (line 167 and on). Instead, the break appears to be where the measurements went from decreasing C_i below 400 ppm to raising it above that C_a level. That instead indicates that the continued exposure to warmer temperatures stimulated photosynthetic processes over the time that the curves were made. Provided that the V_{cmax} data are generated from only low C_i data and the J_{max} data from high C_i data, this may be OK. But there should still be more data points at low C_i - are data points missing?

The reviewer is correct, the low g_s could cause the lower C_i values for a given C_a . We have deleted the text indicated by the reviewer.

The reviewer is also correct in that the continued exposure to warmer temperatures could have stimulated photosynthetic processes especially in the field samples. The procedure for AC_i curve measurement was set to decrease from 400 ppm to 50 ppm, and the warm temperature in the cuvette relative to the field temperatures may have affected

photosynthetic processes by the time the 50ppm measurements were made. In addition, in these measurements the maximum carboxylation rate (V_{cmax}) and maximum electron transport rate (J_{max}) were estimated according to Sharkey *et al.*⁵¹, and as noted by these authors "Data at very low [CO₂] can be limited by Rubisco deactivation and it may be useful to exclude them from the analysis.". With both these issues in mind, we removed the 50ppm data point and only included data from the 100, 150 and 200ppm measurements for our calculations of V_{cmax} . However, for comparison purposes in the table below we compare the calculated values of V_{cmax} including the 50ppm data point with those that we report that did not utilize this measurement point. These data clearly show that including the 50ppm data point does not alter the outcome from the comparisons we report nor our final conclusions. While we could include these data if the Editor wishes us to, we would prefer to not change the V_{cmax} calculations because the seedling data is also calculated from a dataset that does NOT include 50ppm measurements – according to the recommendations of Sharkey *et al.* - and we feel the dataset is more coherent if all model outputs are from datasets using a similar range of CO₂ concentrations.

V_{cmax}	Original-Without		New-With		Sig.	P values
	Mean	S.E.	Mean	S.E.		
Pine-Cold	11.03	3.62	5.39	2.16	NS	0.216838
Pine-recovered	70.84	7.96	77.18	8.53	NS	0.601704
Spruce-Cold	115.45	35.74	112.66	29.39	NS	0.953875
Spruce-recovered	86.42	18.78	85.24	19.81	NS	0.966582

Significance was calculated between the two sets of calculations of V_{cmax} with or without 50 ppm data point. NS, no significance. Two tail T-Test, n=4-6.